# Mapping evidence on factors contributing to maternal and child mortality in sub-Saharan Africa: A scoping review protocol

**Ugochinyere I. Nwagbara**[1]\*, **Emmanuella C. Osuala**[2], **Rumbidzai Chireshe**[1], **Gbotemi B. Babatunde**[3], **Ngozichika O. Okeke**[4], **Nnennaya Opara**[5,6], **Khumbulani W. Hlongwana**[1,7]

**1** Discipline of Public Health Medicine, College of Health Sciences, University of KwaZulu-Natal, Durban, South Africa, **2** Discipline of Pharmaceutical Sciences, College of Health Sciences, University of KwaZulu-Natal, Durban, South Africa, **3** School of Nursing & Public Health, University of KwaZulu-Natal, Durban, South Africa, **4** Department of Nursing, University of Johannesburg, Johannesburg, South Africa, **5** West Virginia School of Osteopathic Medicine, Lewisburg, West Virginia, United States of America, **6** Department of Health Administration, University of Phoenix, Phoenix, Arizona, United States of America, **7** Cancer & Infectious Diseases Epidemiology Research Unit (CIDERU), College of Health Sciences, University of KwaZulu-Natal, Durban, South Africa

\* 216045259@stu.ukzn.ac.za, ugochinyereijeoma@gmail.com

**Data Availability Statement:** Deidentified research data will be made publicly available when the study is completed and published.

## Abstract

### Background

Maternal and child mortality remains a major public health issue in sub-Saharan Africa (SSA), with the region having the highest under-five mortality rates, where approximately 1 in 11 children, dies before the age of 5 years. This is nearly 15 times the average in high-income countries (HICs). This scoping review is aimed at mapping evidence on the factors contributing to maternal and child mortality in SSA.

### Methods

This study will be conducted using a scoping review to map existing literature on the factors contributing to maternal and child mortality in SSA. The search will comprise of peer-reviewed and grey literature, using the EBSCOhost platform. Keyword search from electronic databases such as PubMed/MEDLINE, Google Scholar, Science Direct and World Health Organization library, will be conducted. Information will be obtained from the included studies, using a data charting table. We will use NVIVO version 10 software to analyse the data, and the narrative account of the study will be presented by means of a thematic content analysis.

### Discussion

We expect to find relevant literature that can help us in mapping evidence on the factors contributing to maternal and child mortality in SSA. This study results are anticipated to identify research gaps and in turn, guide the design of future primary studies.

**Funding:** The author(s) received no specific funding for this work.

**Competing interests:** The authors have declared that no competing interests exist.

**Abbreviations: LMICs**, Low-and Middle-Income countries; **MESH**, Medical Subject Headings; **MMAT**, Mixed Method Appraisal Tool; **PCC**, Population Content Context; **PRISMA-P**, Preferred Reporting Items for Systematic Reviews and Meta-Analysis Protocols; **SSA**, sub-Saharan Africa; **UKZN**, University of KwaZulu-Natal; **UN**, United Nations; **WHO**, World Health Organization.

## Systematic review protocol registration

Open Science Framework registration number (DOI 10.17605/OSF.IO/XF5VN).

## Introduction

Maternal and child mortality is a major public health challenge that is among other things, indicative of poor population health and socio-economic development [1]. The World Health Organization (WHO) defined Maternal death as the demise of a woman during pregnancy or within 42 days of terminating a pregnancy from any cause related to the pregnancy or its management [2]. Child mortality is the death of a child under five years of age [3].

Globally, many women die yearly, as a result of complications related to pregnancy, childbearing, and postnatal, with low-and middle-income countries (LMICs) carrying the heaviest burden of maternal mortality [4]. In 2017, two regions of the world, namely: sub-Saharan Africa (SSA) and South Asia, reported about 86% (254,000) of maternal deaths, worldwide, with sub-Saharan Africa and Southern Asia accounting for about 66% (196,000), and 20% (58,000), respectively [2]. Through the two targets of the third Sustainable Development Goal (SDG 3), the United Nations (UN) has committed to reducing maternal and child death worldwide. The goal is that, by 2030, maternal mortality ratio will be decreased to less than 70 per 100,000 live births, globally [5]. This is anticipated to end deaths of newborns and children under 5 years of age, with all countries targeting to reduce neonatal mortality to as low as 12 per 1,000 live births and under-5 mortality to 25 per 1,000 live births [5].

The leading causes of maternal mortality, which accounts for up to 80% of cases in SSA, are obstetric hemorrhage and labor, ruptured uterus, puerperal sepsis, pregnancy-induced hypertension (including eclampsia), and complications of unsafe abortion [6].

Most maternal and infant deaths occur in the first month after birth, and nearly half of postnatal maternal deaths occur within the first 24 hours [7]. Globally, SSA remains the region with the highest child mortality rates, reporting an average under-five mortality rate of 76 deaths per 1,000 live births in 2019, translating into 1 in 13 children dying before reaching the age of 5 years [8].

The leading causes of under-five deaths in SSA are pre-term birth complications (17%), pneumonia (15%), complications during labour and delivery (11%), diarrhoea (9%), and malaria (7%). Under-nutrition contributes to nearly half of all under-five deaths [9]. Despite universal progress in reducing maternal and child mortality, the rates remain high in SSA. Hence immediate action is needed to meet the ambitious SDG 2030 target, which is to eliminate preventable maternal and child mortality ultimately. It is crucial to map out factors contributing to maternal and child deaths in sub-Saharan African countries to inform the design and implementation of appropriate interventions to confront the challenges of the unyielding high rates of mortality in SSA. Thus, it is anticipated that findings from this scoping review on the factors contributing to maternal and child mortality will inform health experts and policymakers in policy development and interventions to bridge the gap in SSA.

### Research question

What is known about the factors contributing to maternal and child mortality in sub-Saharan Africa?

## Materials and methods

### Study design

We will systematically conduct a scoping review of peer-reviewed and grey literature to map evidence on the factors contributing to maternal and child mortality in SSA. We will adopt the scoping review methods proposed by Arksey and O'Malley [10] by following the outlined steps: (i) identifying the research question, (ii) identifying the relevant studies, (iii) study selection, (iv) charting the data, and (v) collating, summarising, and reporting the results. The review protocol has been registered with the Open Science Framework database (registration number: DOI 10.17605/OSF.IO/XF5VN) and will be drafted according to the Preferred Reporting Items for Systematic Reviews and Meta-Analyses Protocols (PRISMA-P) statement [11] **(see Checklist as presented in** S1 File**).**

### Eligibility criteria

We will use the Population-Concept-Context (PCC) framework (Table 1) to determine the eligibility of the research question.

### Inclusion criteria

We will include any original research articles reporting on the factors contributing to maternal and child mortality. Studies published in peer-reviewed journals and conducted in sub-Saharan Africa, using any study design to address maternal and child mortality will be included. Only studies published in English language from January 1990 to date will be considered.

### Exclusion criteria

We will exclude non-English studies as previous studies have shown that it had a minimal effect on the overall conclusions [12,13]. Correspondences, commentaries, editorials, and case reports will be excluded. Studies published before January 1990 will be excluded since the United Nations (UN) world summit for children was conducted in 1990, and it demonstrated the bond between maternal and child health.

### Information sources

To identify relevant studies, the researcher will perform a keyword search with the use of the following electronic databases: PubMed, Google scholar, World Health Organization (WHO) library, EBSCOhost platform (Academic search complete, Health Source: Nursing/Academic Edition, and MEDLINE). We will also search for articles through the "Cited by" search on google scholar, as well as citations in the reference lists of included articles. Our search string will comprise of keywords and Medical Subject Headings (MeSH) terms such as maternal and child, mortality, factors, and sub-Saharan Africa. We have developed a preliminary search

**Table 1. PCC Framework for defining the eligibility of the research question.**

| Criteria | Determinant |
|---|---|
| Population | Female participants ≥15 years of age<br>Children ≤ 5 years of age |
| Concept | Articles focusing on the factors contributing to maternal and child mortality |
| Context | Sub-Saharan Africa |

strategy using PUBMED, presented in S2 File. Boolean terms (AND, OR) will be used to separate the keywords during the search.

## Study selection

A PRISMA flow chart will be used to summarize the study findings (Fig 1) [14]. All the studies meeting the inclusion criteria will be exported to Endnote X7 software. Title and abstract screening will be conducted by two independent reviewers and all the studies that do not address the study's research question will be excluded, along with all the duplicates. Full-text articles will be retrieved for all the abstracts that meet the inclusion criteria for full-text screening. If there are articles that are hard to find, assistance from the University of KwaZulu-Natal (UKZN) library services will be sought by the researchers. Authors will be contacted to request for full-text copies of articles that are not attainable within the UKZN library. A third reviewer will be requested to resolve any discrepancies if the reviewers cannot reach a consensus on some articles during the screening. Full articles of eligible studies will be used for data extraction.

The Preferred Report Items for Systematic and Meta-Analysis (PRISMA) flow chart for the selection and screening of studies is shown in Fig 1.

## Data extraction and data charting

Once the final list of the articles that meet the inclusion criteria has been agreed upon, one reviewer will extract the data relevant to the objectives of the study. The data set will be entered into an Excel form which will be updated continuously. The form will comprise of the following: author and date, title of the study, country, study aims, study design, study setting, population, outcomes of the study, and key findings (see S3 File).

## Collating, summarising, and reporting the results

The findings from the included studies will be presented through a thematic content analysis approach. Data will be retrieved for the following outcomes: maternal and child mortality, and factors contributing to maternal and child mortality. Other emerging themes will be analysed and examined to determine whether the themes answer the research question. The significance of the results with the overall aim of the study and the urgent need for advanced research, new policy, and improved health practices will be discussed by the review team. NVIVO version 10 software will be used to code the data from the included studies [15]. The following process will serve as guidelines:

- Data coding

- Categorisation of the codes into major themes

- Data presentation

- Identification of key patterns and sub-themes.

- Summarising

## Synthesis

The retrieved data obtained from the selected research articles will be examined for emerging themes related to the research questions. The correlation of the findings with the aim of the study and relevance of these findings for future research, new health policies, and health promotion practices will be examined by the reviewers.

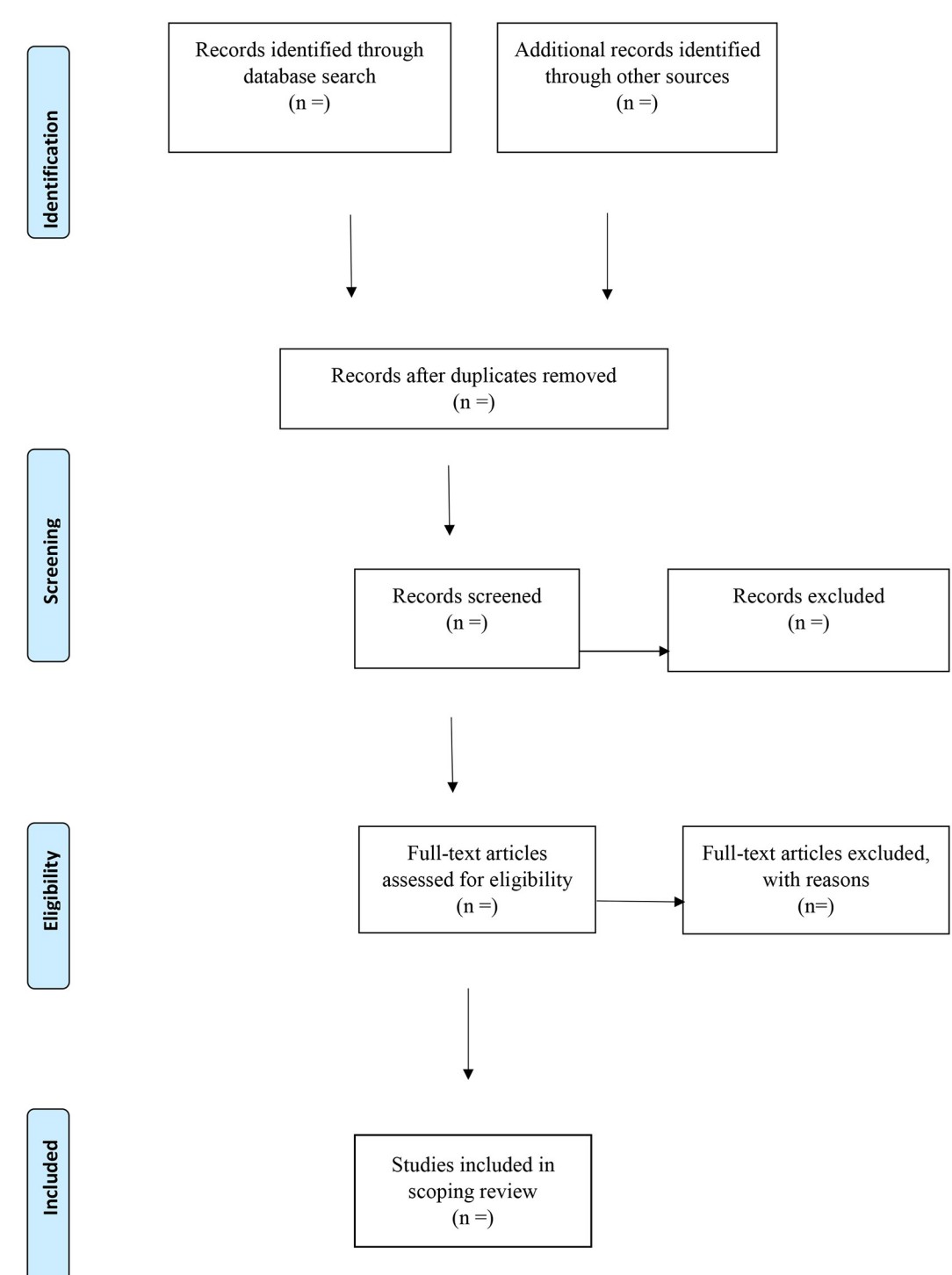

**Fig 1. PRISMA flow-chart.** [Source: Adapted from Moher et al. [14]].

## Quality appraisal

Quality assessment of selected articles will not be performed in accordance with scoping review methodology. Our study aims to map appropriate literature and identify evidence on the significant factors contributing to maternal and child mortality [10,16].

## Discussion

This scoping review findings will reveal gaps and form the basis for refining research questions for future studies. At the time of writing this manuscript, we could not find any similar scoping reviews published on the same issue.

This protocol is designed to outline the plans for the scoping review. The search strategy of various databases and grey literatures as described in this protocol will ensure a thorough literature search to reduce bias.

The review findings will be published in a peer-reviewed journal upon completion of the study. The findings from this review are expected to raise awareness among policymakers on the urgency for newer and more effective policies and guidelines aimed at reducing maternal and infant mortality. Furthermore, the identified gaps will guide researchers in the design of future primary studies.

## Supporting information

**S1 Checklist. PRISMA-P 2015 checklist.**
(DOCX)

**S1 File. Preferred reporting items for systematic reviews and meta-analysis protocols (PRISMA-P) checklist.**
(PDF)

**S2 File. PubMed search strategy.**
(DOCX)

**S3 File. Data charting form.**
(DOCX)

## Author Contributions

**Conceptualization:** Ugochinyere I. Nwagbara, Emmanuella C. Osuala, Rumbidzai Chireshe, Gbotemi B. Babatunde, Ngozichika O. Okeke, Nnennaya Opara, Khumbulani W. Hlongwana.

**Data curation:** Ugochinyere I. Nwagbara, Emmanuella C. Osuala, Rumbidzai Chireshe, Gbotemi B. Babatunde, Khumbulani W. Hlongwana.

**Formal analysis:** Ugochinyere I. Nwagbara, Emmanuella C. Osuala, Gbotemi B. Babatunde.

**Investigation:** Ugochinyere I. Nwagbara, Emmanuella C. Osuala, Rumbidzai Chireshe, Gbotemi B. Babatunde.

**Methodology:** Ugochinyere I. Nwagbara, Emmanuella C. Osuala, Rumbidzai Chireshe, Gbotemi B. Babatunde, Khumbulani W. Hlongwana.

**Supervision:** Khumbulani W. Hlongwana.

**Validation:** Ugochinyere I. Nwagbara, Emmanuella C. Osuala, Rumbidzai Chireshe, Gbotemi B. Babatunde, Ngozichika O. Okeke, Nnennaya Opara, Khumbulani W. Hlongwana.

**Writing – original draft:** Ugochinyere I. Nwagbara, Emmanuella C. Osuala, Rumbidzai Chireshe, Gbotemi B. Babatunde, Ngozichika O. Okeke, Nnennaya Opara, Khumbulani W. Hlongwana.

**Writing – review & editing:** Ugochinyere I. Nwagbara, Emmanuella C. Osuala, Rumbidzai Chireshe, Gbotemi B. Babatunde, Ngozichika O. Okeke, Nnennaya Opara, Khumbulani W. Hlongwana.

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
