## [Decision Letter · Decision Letter 0]

21 Apr 2022

PONE-D-21-16423

Mapping evidence on factors contributing to maternal and child mortality in sub-Saharan Africa: a scoping review protocol

PLOS ONE

Dear Dr.  Nwagbara et al,

Thank you for submitting your manuscript to PLOS ONE. After careful consideration, we feel that it has merit but does not fully meet PLOS ONE’s publication criteria as it currently stands. Therefore, we invite you to submit a revised version of the manuscript that addresses the points raised during the review process.

It is okay to not include studies published in French, however, you need to justify this.

We look forward to receiving your revised manuscript.

Kind regards,

Akanni Ibukun Akinyemi, PhD

Academic Editor

PLOS ONE

Journal Requirements:

3. Thank you for submitting the above manuscript to PLOS ONE. During our internal evaluation of the manuscript, we found significant text overlap between your submission and the following previously published works, some of which you are an author.

-   https://www.researchsquare.com/article/rs-104159/v1

-   https://www.researchsquare.com/article/rs-6092/v1

-   https://www.ncbi.nlm.nih.gov/books/NBK2288/

-   https://ourworldindata.org/child-mortality

-   https://www.who.int./mediacentre/news/releases/2014/child_mortality_estimates/en/

Please revise the manuscript to rephrase the duplicated text, cite your sources, and provide details as to how the current manuscript advances on previous work. Please note that further consideration is dependent on the submission of a manuscript that addresses these concerns about the overlap in text with published work.

Reviewers' comments:

Reviewer's Responses to Questions

**Comments to the Author**

1. Does the manuscript provide a valid rationale for the proposed study, with clearly identified and justified research questions?

Reviewer #1: Yes

Reviewer #2: Partly

2. Is the protocol technically sound and planned in a manner that will lead to a meaningful outcome and allow testing the stated hypotheses?

Reviewer #1: Yes

Reviewer #2: No

3. Is the methodology feasible and described in sufficient detail to allow the work to be replicable?

Reviewer #1: Yes

Reviewer #2: Yes

4. Have the authors described where all data underlying the findings will be made available when the study is complete?

Reviewer #1: Yes

Reviewer #2: Yes

5. Is the manuscript presented in an intelligible fashion and written in standard English?

Reviewer #1: Yes

Reviewer #2: Yes

6. Review Comments to the Author

You may also provide optional suggestions and comments to authors that they might find helpful in planning their study.

Reviewer #1: Dear Author, Please consider below suggestions:

1. In the introduction you have given the definition of maternal death. I suggest you give it earlier in the introduction section.

2. In the research question, actually you are searching for the gap of knowledge, and then please mention it properly and mention any potential sub-questions.

3. I do not agree with this type of writing inclusion and exclusion criteria. Actually, it is a kind of repeating the same thing.

4. Please mention the approval of the protocol and funding source.

Reviewer #2: Many thanks for the opportunity to review the manuscript Mapping evidence on factors contributing to maternal and child mortality in sub-Saharan Africa: a scoping review protocol.

I think the authors have develop a protocol which to certain extent meets the requirement for a scoping review protocol.

I am missing the search strategy for all the databases which should be in an appendix. I also think that the suggested PubMed search is not build to capture factors contributing to maternal and child mortality e.g. I don’t see the word cause of death etc.

I don’t understand why you are not including studies published in French as SSA has a huge Francophone population.

For the data extraction – what do the authors mean with “most relevant findings”?

For the reviewer it would be very useful if the manuscript has line numbering.

Why women older than 15 years?

The references used are generally very old e.g. World Health Statistics from 2016. A new report is published very year. The same with reference 9.

I don’t think this review adds much to the literature as we already know why women and children are dying in SSA.

7. PLOS authors have the option to publish the peer review history of their article (what does this mean?). If published, this will include your full peer review and any attached files.

Reviewer #1: No

Reviewer #2: **Yes: **Ann-Beth Moller

---

## [Author Response · Author response to Decision Letter 0]

5 Jul 2022

Author’s response to the reviews.

Manuscript title: Mapping evidence on factors contributing to maternal and child mortality in sub-Saharan Africa: a scoping review protocol

Manuscript PONE-D-21-16423

We are very grateful for the reviews provided by the editor and the reviewers of this manuscript. The comments were very useful. We have revised the manuscript according to the reviewer’s comments. Below is the point-by-point response to the reviewer’s comments. Please find attached a revised version of the manuscript with tracked changes highlighted in ‘red’.

Response to reviewers’ queries/ comments

EDITOR’S COMMENTS

# Editor’s comment Authors’ responses Page and Line Number/s

1 It is okay to not include studies published in French, however, you need to justify this. We agree with the reviewer that the justification should have been provided, hence we have now provided it. In short, we will exclude non-English studies as previous studies have shown that it had a minimal effect on the overall conclusions.

 Page 6, Lines 114-115

2 Please ensure that your manuscript meets PLOS ONE's style requirements, including those for file naming. The PLOS ONE style templates can be found at 

https://journals.plos.org/plosone/s/fil es can be found at 

 We have reviewed the style requirements again and we are of the view that our manuscript is complaint Whole Manuscript

3 In your Data Availability statement, you have not specified where the minimal data set underlying the results described in your manuscript can be found. PLOS defines a study's minimal data set as the underlying data used to reach the conclusions drawn in the manuscript and any additional data required to replicate the reported study findings in their entirety. All PLOS journals require that the minimal data set be made fully available. For more information about our data policy, please see http://journals.plos.org/plosone/s/data-availability.

Important: If there are ethical or legal restrictions to sharing your data publicly, please explain these restrictions in detail. Please see our guidelines for more information on what we consider unacceptable restrictions to publicly sharing data: http://journals.plos.org/plosone/s/data-availability#loc-unacceptable-data-access-restrictions. Note that it is not acceptable for the authors to be the sole named individuals responsible for ensuring data access. We have added our data Availability statement Page 11, Lines 254-255

4 3. Thank you for submitting the above manuscript to PLOS ONE. During our internal evaluation of the manuscript, we found significant text overlap between your submission and the following previously published works, some of which you are an author.

- https://www.researchsquare.com/article/rs-104159/v1

- https://www.researchsquare.com/article/rs-6092/v1

- https://www.ncbi.nlm.nih.gov/books/NBK2288/

- https://ourworldindata.org/child-mortality

- https://www.who.int./mediacentre/news/releases/2014/child_mortality_estimates/en/

Please revise the manuscript to rephrase the duplicated text, cite your sources, and provide details as to how the current manuscript advances on previous work. Please note that further consideration is dependent on the submission of a manuscript that addresses these concerns about the overlap in text with published work.

We will carefully review your manuscript upon resubmission, so please ensure that your revision is thorough. Thanks for the notification as we have carefully rephrased the duplicated texts Whole Manuscript

REVIEWER ONE COMMENT

1 Reviewer #1: Dear Author, Please consider below suggestions:

In the introduction you have given the definition of maternal death. I suggest you give it earlier in the introduction section. Thank you for this good suggestion, we have now moved the definition of maternal death to the earlier part of the introduction, as suggested. Page 3, Lines 49-52

2 In the research question, actually you are searching for the gap of knowledge, and then please mention it properly and mention any potential sub-questions. Thank you for this suggestion, we have now mentioned the gap of knowledge and potential sub-questions properly. Page 4, Lines 85-86

3 I do not agree with this type of writing inclusion and exclusion criteria. Actually, it is a kind of repeating the same thing. Thanks for the observation. We have re-written our inclusion and exclusion criteria in an unrepetitive manner. Pages 5-6, Lines 107-119

4 Please mention the approval of the protocol and funding source. The requested information has now been incorporated, thanks. Page 11, Line 257

REVIEWER TWO COMMENT

1 Reviewers' comments:

Many thanks for the opportunity to review the manuscript Mapping evidence on factors contributing to maternal and child mortality in sub-Saharan Africa: a scoping review protocol.

I think the authors have develop a protocol which to certain extent meets the requirement for a scoping review protocol.

 Thank for your review and comment. 

2 I am missing the search strategy for all the databases which should be in an appendix. I also think that the suggested PubMed search is not build to capture factors contributing to maternal and child mortality e.g. I don’t see the word cause of death etc. Thanks for your observation. We have added a pilot search using the PubMed database and also included the word ‘cause of death’ in the search. Page 10-Lines 238-239

3 I don’t understand why you are not including studies published in French as SSA has a huge Francophone population. We have taken an approach by Reviewer One, suggesting that excluding non-English articles is acceptable, provided that proper justification is provided (please see our response to the related query by Reviewer One).

 Page 6, Lines 114-115

4 For the data extraction – what do the authors mean with “most relevant findings”? Upon reflection, we have released that this phrase ‘most relevant findings’ may be confusing, hence we have removed it from our data extraction form and replaced it with ‘key findings’. During the data extraction process we will include the key outcomes from each article included in the study. Page 10, Line 240

5 For the reviewer it would be very useful if the manuscript has line numbering We have inserted line numbering. Whole Manuscript

6 Why women older than 15 years? We chose this age due to the high prevalence of teenage pregnancy in SSA in this age group. The categorisation of 15-24 years or 15-19 years (in the case of studies on teenage issues) is common in scientific papers. So, we expect that most studies reporting on maternal and child health issues may rarely capture ages below 15 years. 

7 The references used are generally very old e.g. World Health Statistics from 2016. A new report is published very year. The same with reference 9. Thanks, we have updated the references. Whole Manuscript

8 I don’t think this review adds much to the literature as we already know why women and children are dying in SSA.

 This study is trying to challenge the very dismissive perception that we know the problems in SSA and not be opened to observing the emerging trends. Greater emphasis still needs to be paid to maternal and child health issues, as these remain serious public health problems. Unless, this maternal and child health issues remain in the agenda of scientists, policy makers, advocacy groups and interventionists may easily lose sight to these important issues. .

---

## [Editor Report · Decision Letter 1]

19 Jul 2022

Mapping evidence on factors contributing to maternal and child mortality in sub-Saharan Africa: a scoping review protocol

PONE-D-21-16423R1

Dear Dr. Nwagbara,

We’re pleased to inform you that your manuscript has been judged scientifically suitable for publication and will be formally accepted for publication once it meets all outstanding technical requirements.

Kind regards,

Akanni Ibukun Akinyemi, PhD

Academic Editor

PLOS ONE
---

## [Editor Report · Acceptance letter]

29 Jul 2022

PONE-D-21-16423R1 

Mapping evidence on factors contributing to maternal and child mortality in sub-Saharan Africa: a scoping review protocol 

Dear Dr. Nwagbara:

I'm pleased to inform you that your manuscript has been deemed suitable for publication in PLOS ONE. Congratulations! Your manuscript is now with our production department. 

Kind regards, 

on behalf of

Dr. Akanni Ibukun Akinyemi 

Academic Editor

PLOS ONE